# On the Sensitivity of Adversarial Robustness to Input Data Distributions

**Gavin Weiguang Ding, Kry Yik Chau Lui, Xiaomeng Jin, Luyu Wang, Ruitong Huang**
Borealis AI
Canada

## Abstract

Neural networks are vulnerable to small adversarial perturbations. Existing literature largely focused on understanding and mitigating the vulnerability of learned *models*. In this paper, we demonstrate an intriguing phenomenon about the most popular robust training method in the literature, adversarial training: Adversarial robustness, unlike clean accuracy, is sensitive to the *input data distribution*. Even a semantics-preserving transformations on the input data distribution can cause a significantly different robustness for the adversarial trained model that is both trained and evaluated on the new distribution. Our discovery of such sensitivity on data distribution is based on a study which disentangles the behaviors of clean accuracy and robust accuracy of the Bayes classifier. Empirical investigations further confirm our finding. We construct semantically-identical variants for MNIST and CIFAR10 respectively, and show that standardly trained models achieve comparable clean accuracies on them, but adversarially trained models achieve significantly different robustness accuracies. This counter-intuitive phenomenon indicates that input data distribution alone can affect the adversarial robustness of trained neural networks, not necessarily the tasks themselves. Lastly, we discuss the practical implications on evaluating adversarial robustness, and make initial attempts to understand this complex phenomenon.

## 1 Introduction

Neural networks have been demonstrated to be vulnerable to adversarial examples (Szegedy et al., 2013; Biggio et al., 2013). Since the first discovery of adversarial examples, great progress has been made in constructing stronger adversarial attacks (Goodfellow et al., 2014; Moosavi-Dezfooli et al., 2016; Madry et al., 2017; Carlini and Wagner, 2017). In contrast, defenses fell behind in the arms race (Carlini and Wagner, 2016; Athalye et al., 2017; 2018). Recently a line of works have been focusing on understanding the difficulty in achieving adversarial robustness from the perspective of data distribution. In particular, Tsipras et al. (2019) demonstrated the inevitable tradeoff between robustness and clean accuracy in some particular examples. Schmidt et al. (2018) showed that the sample complexity of "learning to be robust" learning could be significantly higher than that of "learning to be accurate".

In this paper, we contribute to this growing literature from a new angle, by studying the relationship between adversarial robustness and the input data distribution. We focus on the adversarial training method, arguably the most popular defense method so far due to its simplicity, effectiveness and scalability (Goodfellow et al., 2014; Huang et al., 2015; Kurakin et al., 2016; Madry et al., 2017; Erraqabi et al., 2018). Our main contribution is the finding that adversarial robustness is highly sensitive to the input data distribution:

*A semantically-lossless shift on the data distribution could result in a drastically different robustness for adversarially trained models.*

Note that this is different from the transferability of a fixed model that is trained on one data distribution but tested on another distribution. Even retraining the model on the new data distribution may give us a completely different adversarial robustness on the same new distribution. This is also in sharp contrast to the clean accuracy of standard training, which, as we show in later sections, is insensitive to such shifts. To our best knowledge, our paper is the first work in the literature that demonstrates such sensitivity.

Our investigation is motivated by the empirical observations on the MNIST dataset and the CIFAR10 dataset. In particular, while comparable SOTA clean accuracies (the difference is less than 3%) are achieved by MNIST and CIFAR10 (Gastaldi, 2017), CIFAR10 suffers from much lower achievable robustness than MNIST in practice.[1] Results of this paper consist of two parts. First in theory, we start with analyzing the difference between the regular Bayes error and the robust error, and show that the regular Bayes error is invariant to invertible transformations of the data distribution, but the robust error is not. We further prove that if the input data is uniformly distributed, then the perfect decision boundary cannot be robust. However, we also manage to find a robust model for the binarized MNIST dataset (semantically almost identical to MNIST, later described in Section 3). The certification method by Wong and Kolter (2018) guarantees that this model achieves at most 3% robust error. Such a sharp contrast suggests the important role of the data distribution in adversarial robustness, and leads to our second contribution on the empirical side: we design a series of augmented MNIST and CIFAR10 datasets to demonstrate the sensitivity of adversarial robustness to the input data distribution.

Our finding of such sensitivity raises the question of how to properly evaluate adversarial robustness. In particular, the sensitivity of adversarial robustness suggests that certain datasets may not be sufficiently representative when benchmarking different robust learning algorithms. It also raises serious concerns about the deployment of believed-to-be-robust training algorithm in a real product. In a standard development procedure, various models (for example different network architectures) would be prototyped and measured on the existing data. However, the sensitivity of adversarial robustness makes the truthfulness of the performance estimations questionable, as one would expect future data to be slightly shifted. We illustrate the practical implications in Section 4 with two practical examples: 1) the robust accuracy of PGD trained model is sensitive to gamma values of gamma-corrected CIFAR10 images. This indicates that image datasets collected under different light conditions may have different robustness properties; 2) both as a "harder" version of MNIST, the fashion-MNIST (Xiao et al., 2017) and edge-fashion-MNIST (an edge detection variant described in Section 4.2) exhibit completely different robustness characteristics. This demonstrates that different datasets may give completely different evaluations for the same algorithm.

Finally, our finding opens up a new angle and provides novel insights to the adversarial vulnerability problem, complementing several recent works on the issue of data distributions' influences on robustness. Tsipras et al. (2019) hypothesize that there is an intrinsic tradeoff between clean accuracy and adversarial robustness. Our studies complement this result, showing that there are different levels of tradeoffs depending on the characteristics of input data distribution, under the same learning settings (training algorithm, model and training set size). Schmidt et al. (2018) show that different data distributions could have drastically different properties of adversarially robust generalization, theoretically on Bernoulli vs mixtures of Gaussians, and empirically on standard benchmark datasets. From the sensitivity perspective, we demonstrate that being from completely different distributions (e.g. binary vs Gaussian or MNIST vs CIFAR10) may not be the essential reason for having large robustness difference. Gradual semantics-preserving transformations of data distribution can also cause large changes to datasets' achievable robustness. We make initial attempts in Section 5 to further understand this sensitivity. We investigated perturbable volume and inter-class distance as the natural causes of the sensitivity; model capacity and sample complexity as the natural remedies. However, the complexity of the problem has so far defied our efforts to give a definitive answer.

## 1.1 NOTATION AND PROBLEM SETUP

We specifically consider the image classification problem where the input data is inside a high dimensional unit cube. We denote the data distribution as a joint distribution $\mathbb{P}(x, y)$, where $x \in [0, 1]^d$, $d$ is the number of pixels, and $y \in \{1, 2, \ldots, k\}$ is the discrete label. We assume the support of $x$ is the whole pixel space $[0, 1]^d$. When $x$ is a random noise (or human perceptually unclassifiable image), one can think of $\mathbb{P}(y \mid x)$ being closed to uniform distribution on labels. In the standard setting, the samples $(x_i, y_i)$ can be interpreted as $x_i$ is independently sampled from the marginal distribution $\mathbb{P}(x)$, and then $y_i$ is sampled from $\mathbb{P}(x \mid x_i)$. In this paper, we discuss $\mathbb{P}(x)$'s influences on adversarial robustness, given a fixed $\mathbb{P}(y|x)$.

---

[1]After PGD adversarial training (Madry et al., 2017), MNIST has 89.3% accuracy under perturbation $\delta$ with $\|\delta\|_\infty \leq 0.3$ under the strongest PGD attack, but CIFAR10 has only 45.8% accuracy, even under a much smaller $\delta$ with $\|\delta\|_\infty \leq 8/255$.

In our experiments, we only discuss the whitebox robustness, as it represents the "intrinsic" robustness. We use models learned by adversarially augmented training (Madry et al., 2017) (PGD training), which has the SOTA whitebox robustness. We consider bounded $\ell_\infty$ attack as the attack for evaluating robustness for 2 reasons: 1) PGD training can defend against $\ell_\infty$ relatively well, while for other attacks, how to train a robust model is still an open question; 2) in the image domain $\ell_\infty$ attack is the mostly widely researched attack.

Let $\mathcal{H}$ denote the universal set of all the measurable functions. Given a joint distribution $\mathbb{P}(x, y)$ on the space $\mathcal{X} \times \mathcal{Y}$, we define the Bayes error $R^* = \inf_{h \in \mathcal{H}} \mathbb{E}_{\mathbb{P}(x,y)} L(y; h(x)) = R^*(\mathbb{P}(x, y))$, where $L$ is the objective function. In other words, Bayes error is the error of the best possible classifier we can have, $h^*$, without restriction on the function space of classifiers. We further define (adversarial) robust error $RR(h) = \mathbb{E}_{\mathbb{P}(x,y)} \max_{\|\delta\|_\infty < \epsilon} L(y; h(x + \delta)) = RR(\mathbb{P}(x, y))$. We denote $RR^* = RR(h^*)$ to be the robust error achieved by the Bayes classifier $h^*$. For simplicity, we assume our algorithm can always learn $h^*$, which reduces clean accuracy to be $(1 - \text{Bayes error})$, and robust accuracy of the Bayes classifier to be $(1 - RR^*)$.

## 2 THEORETICAL ANALYSES AND PROVABLE CASES

As mentioned in the introduction, although the SOTA clean accuracies are similar for MNIST and CIFAR10, the robust accuracy on CIFAR10 is much more difficult to achieve, which indicates the different behaviors of the clean accuracy and robust accuracy. The first result in this section is to further confirm this indication in a simple setting, where the clean accuracy remains the same but the robust accuracy completely changes under a distribution shift. Based on results from the concentration of measure literature, we further show that under uniform distribution, no algorithm can achieve good robustness, as long as they have high clean accuracy. On the other hand, we examine the performance of a verifiable defense method on binarized MNIST (pixels values rounded to 0 and 1), and the result suggests the exact opposite: provable adversarial robustness on a MNIST-like dataset is achievable. Such contrast thus suggests the important role of the data distribution in achieving adversarial robustness.

### 2.1 DISENTANGLE CLEAN ACCURACY AND ROBUST ACCURACY

One immediate result is that Bayes error remains the same under any distribution shift induced by an injective map $T : \mathcal{X} \to \mathcal{X}$. To see that, simply note that $T^{-1}$ exists and $h^* \circ T^{-1}$ gives the same Bayes error for the shifted distribution. However, such invariance property does not hold for the robust error of the Bayes classifier. Furthermore, the following two examples show that Bayes error can have completely different behavior from its robust error. Although both examples have 0 Bayes error, they have completely different robust errors.

**Example 1.** *Assume $x$ is uniformly distributed in $[0, 1]^d$ and $y = 1$, for all $x$ with $x^\top e_1 > 1/2$ and $y = 0$, for $x^\top e_1 \leq 1/2$, where $e_1$ is the one-hot vector. We use the 0-1 loss here. Note that the Bayes error decision boundaries are given by the following hyperplane: $HP_1 = \{x \in [0, 1]^d : x_1 = 0\}$, and thus*

$$R^* = 0; \qquad RR^* = 2\epsilon,$$

*under the budget $\|\delta\|_\infty < \epsilon$. In this case, the robust error is tolerable and relatively robust measured by the fraction of points that are successfully attacked, $2\epsilon$.*

*Moreover, consider an injective map $T$ which maps $\{x : x^\top e_1 > 1/2\}$ to $\{x : x^\top \boldsymbol{1} > \frac{d}{2}\}$, and $\{x : x^\top e_1 \leq 1/2\}$ to $\{x : x^\top \boldsymbol{1} \leq \frac{d}{2}\}$[2]. The Bayes error on the new distribution remains 0, as $T$ is invertible. In contrast, the robust error is much worse. In fact,*

$$RR^* \geq 1 - \frac{1}{4d\epsilon^2}.$$

**Remark 2.1.** *Note that here the robust error of the Bayes classifier will grow to 1 as the dimensionality increases, for a fixed budget $\epsilon$.*

### 2.2 DIFFICULTY IN ACHIEVING ROBUSTNESS

Example 1 shows that good clean accuracy does not necessary lead to good robust accuracy. In contrast, we will show in this section that achieving a good robust accuracy is impossible given uniformly distributed data, as long as we ask for good clean accuracies. Our tool are classical results from the concentration of measure (Ledoux, 2005).

---

[2]Such map can be easily constructed.

Let $A_\epsilon := \{x \in \mathbb{R}^N | d(x, A) < \epsilon\}$ denote the $\epsilon$-neighborhood of the nonempty set $A$, where $d(x, A)$ is the distance from $x$ to the set $A$. Theorem 2.1 provides a lower bound on the mass in $A_\epsilon$.

**Theorem 2.1** (Concentration of Measure on the Unit Cube and the Unit Ball). *Let $[0, 1]^d$ denote the unit $d$-cube and $B^d$ denote the Euclidean unit $d$-ball, both equipped with uniform probability distributions. Let $\epsilon > 0$. Then for any $A \subset [0, 1]^d$ with $\mathbb{P}(A) \geq 1/2$, we have:*

$$\mathbb{P}(A_\epsilon) \geq \Phi(\epsilon\sqrt{2\pi} + \Phi^{-1}(\mathbb{P}(A))) \geq 1 - e^{-\pi\epsilon^2} \tag{1}$$

*For any $B \subset B^d$, with $\mathbb{P}(B) \geq 1/2$,*

$$\mathbb{P}(B_\epsilon) \geq 1 - \frac{1}{\mathbb{P}(B)}(1 - \delta_{\ell_2}(\epsilon))^{2d} \geq 1 - \frac{1}{\mathbb{P}(B)}e^{-2d(\frac{2-\sqrt{3}}{3})\epsilon^2} \tag{2}$$

*where $\delta_{\ell_2}(\epsilon) = 1 - \sqrt{1 - \frac{\epsilon^2}{4}}$ and $\Phi$ is the standard normal cumulative distribution function.*

Based on Theorem 2.1 we can now show that under some circumstances, *no* algorithm that achieves can perfect clean accuracy can also achieve a good robust accuracy.

**Example 2** (Vulnerability Guarantee). *Consider the joint distribution $\mathbb{P}(x, y)$, where the input data $x$ is uniformly distributed on $[0, 1]^d$ and label $y$ has 10 classes. Further assume the marginal distribution of $y$ is also uniform[3]. Theorem 2.1 implies that under $\ell_2$ adversarial attack with $\epsilon = 0.5$, at least 94 % of the samples are ether wrongly classified or can be successfully attacked for a classifier with perfect clean accuracy.*

*Furthermore, if $d = 3 \times 32 \times 32$, A parallel calculation for $\mathbb{P}(x, y)$ on the $B^d$ domain gives: under $\ell_2$ adversarial attack with $\epsilon = 0.09$ , at least 97 % of the the samples are ether wrongly classified or can be successfully attacked for a classifier with perfect clean accuracy.*

On the one hand, Theorem 2.1 and Example 2 suggest that the uniform distribution on $[0, 1]^d$ enjoys more robustness than the uniform distribution on $B^d$, and it is not affected by the high dimensionality. This may partially explain why MNIST is more adversarially robust than CIFAR10, as the distribution of $x$ in CIFAR10 is "closer" to $B^d$ than to $[0, 1]^d$. On the other hand, while not completely sharp, they also suggest the intrinsic difficulty in achieving good robust accuracy.

Note that one limit of Theorem 2.1 and Example 2 is the uniform distribution assumption, which is surely not true for natural images. Indeed, although rigorously developed, Theorem 2.1 and Example 2 do *not* explain certain empirical observations. Following Wong and Kolter (2018), we train a provably[4] robust model on a binarized MNIST dataset (bMNIST) [5]. Our experiments shows that the learned model achieves 3.00% provably robust error on bMNIST test data, while maintaining 97.65% clean accuracy. Details of this experiment in described in Appendix B.2.

The above MNIST experiment and Example 2 suggest the essential role of the data distribution in achieving good robust and clean accuracies. While it is hard to completely answer the question what geometric properties differentiate the concentration rates between the ball/cube in high dimension and the distribution of bMNIST, we remark that one obvious difference is the distance distributions in both spaces. Could the distance distributions explain the differences in clean and robust accuracies? Note that the same method can only achieve 37.70% robust error on original MNIST data, and even higher error on CIFAR10, which further supports this hypothesis. In the rest of this paper, we further investigate the dependence of robust accuracy on the distribution of real data.

## 3 ROBUSTNESS ON DATASETS VARIANTS WITH DIFFERENT INPUT DISTRIBUTIONS

Section 2.2 clearly suggests that the data distribution plays an essential role in the achievable robust accuracy. In this section we carefully design a series of datasets and experiments to further study its influence. One important property of our new datasets is that they have different $\mathbb{P}(x)$'s while keep $\mathbb{P}(y|x)$ reasonably fixed, thus these datasets are only different in a "semantic-lossless" shift. Our experiments reveal an unexpected phenomenon that while standard learning methods manage to achieve stable clean accuracies across different data distributions under "semantic-lossless" shifts, however, adversarial training, arguably the most popular method to achieve robust models, loses this

---

[3]but their joint distribution is not necessary uniform.

[4]"Provably" means that the robust accuracy of the model can be rigorously proved.

[5]It is created by rounding all pixel values to 0 or 1 from the original MNIST

desirable property, in that its robust accuracy becomes unstable even under a "semantic-lossless" shift on the data distribution.

We emphasize that different from preprocessing steps or transfer learning, here we treat the shifted data distribution as a new underlying distribution. We both train the models and test the robust accuracies on the same new distribution.

## 3.1 SMOOTHING AND SATURATION

We now explain how the new datasets are generated under "semantic-lossless" shifts. In general, MNIST has a more binary distribution of pixels, while CIFAR10 has a more continuous spectrum of pixel values, as shown in Figure 1a and 1b. To bridge the gap between these two datasets that have completely different robust accuracies, we propose two operations to modify their distribution on $x$: smoothing and saturation, as described below. We apply different levels of "smoothing" on MNIST to create more CIFAR-like datasets, and different levels of "saturation" on CIFAR10 to create more "binary" ones. Note that we would like to maintain the semantic information of the original data, which means that such operations should be semantics-lossless and not arbitrarily wide.

**Smoothing** is applied on MNIST images, to make images "less binary". Given an image $x_i$, its smoothed version $\tilde{x}_i^{(s)}$ is generated by first applying average filter of kernel size $s$ to $x_i$ to generate an intermediate smooth image, and then take pixel-wise maximum between $x_i$ and the intermediate smooth image. Our MNIST variants include the binarized MNIST and smoothed MNIST with different kernel sizes. As shown in Figure 1c, all MNIST variants still maintain the semantic information in MNIST, which indicates that $\mathbb{P}(y \mid \tilde{x}^{(s)})$ should be similar to $\mathbb{P}(y \mid x)$. It is thus reasonable to assume that $y_i$ is approximately sampled from $\mathbb{P}(y \mid \tilde{x}^{(s)})$, and as such we assign $y_i$ as the label of $\tilde{x}^{(s)}$. Note that all the data points in the binarized MNIST are on the corners of the unit cube. For the smoothed versions, pixels on the digit boundaries are pushed off the corner of the unit cube.

**Saturation** of the image $x$ is denoted by $\hat{x}^{(p)}$, and the procedure is defined as below:

$$\hat{x}^{(p)} = \text{sign}(2x - 1)\frac{|2x - 1|^{\frac{2}{p}}}{2} + \frac{1}{2},$$

where all the operations are pixel-wise and each element of $\hat{x}^{(p)}$ is guaranteed to be in $[0, 1]$. Saturation is used to generate variants of the CIFAR10 dataset with less centered pixel values. For different saturation level $p$'s, one can see from Figure 1d that $\hat{x}^{(p)}$ is still semantically similar to $x$ in the same classification task. Similarly we assign $y_i$ as the label of $\hat{x}_i^{(p)}$. One immediate property about $\hat{x}^{(p)}$ is that it pushes $x$ to the corners of the data domain where the pixel values are either 0 or 1 when $p \geq 2$, and pull the data to the center of 0.5 when $p \leq 2$. When $p = 2$ it does not change the image, and when $p = \infty$ it becomes binarization.

## 3.2 EXPERIMENTAL SETUPS

In this section we use the smoothing and saturation operations to manipulate the data distributions of MNIST and CIFAR10, and show empirical results on how data distributions affects robust accuracies of neural networks trained on them. Since we are only concerned with the intrinsic robustness of neural networks models, we do not consider methods like preprocessing that tries to remove perturbations or randomizing inputs. We perform standard neural network training on clean data to measure the difficulty of the classification task, and projected gradient descent (PGD) based adversarial training (Madry et al., 2017) to measure the difficulty to achieve robustness.

By default, we use LeNet5 on all the MNIST variants, and use wide residual networks (Zagoruyko and Komodakis, 2016) with widen factor 4 for all the CIFAR10 variants. Unless otherwise specified, PGD training on MNIST variants and CIFAR10 variants all follows the settings in Madry et al. (2017). Details of network structures and training hyperparameters can be found in Appendix B.

We evaluate the classification performance using the test *accuracy* of standardly trained models on clean unperturbed examples, and the robustness using the *robust accuracy* of PGD trained model, which is the accuracy on adversarially perturbed examples. Although not directly indicating robustness, we report the clean accuracy on PGD trained models to indicate the tradeoff between being accurate and robust. To understand whether low robust accuracy is due to low clean accuracy or vulnerability of model, we also report *robustness w.r.t. predictions*, where the attack is used to perturb against the model's clean prediction, instead of the true label. We use $\ell_\infty$ untargeted PGD attacks (Madry et al., 2017) as our adversary, since it is the strongest attack in general based on our

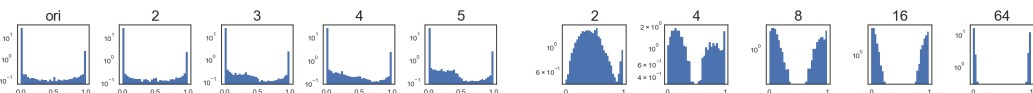

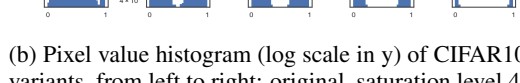

(a) Pixel value histogram (log scale in y) of MNIST variants, from left to right: original, smoothed with kernel size 2, 3, 4, 5

(b) Pixel value histogram (log scale in y) of CIFAR10 variants, from left to right: original, saturation level 4, 8, 16, 64

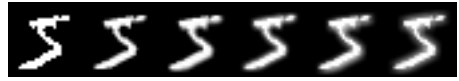

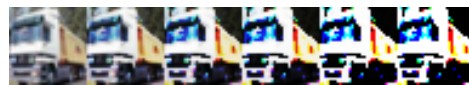

(c) MNIST variants, from left to right: binarized, original, smoothed with kernel size 2, 3, 4, 5

(d) CIFAR10 variants, from left to right, original, saturation level 4, 8, 16, 64, $\infty$

Figure 1: Variants of smoothed MNIST and saturated CIFAR10 datasets.

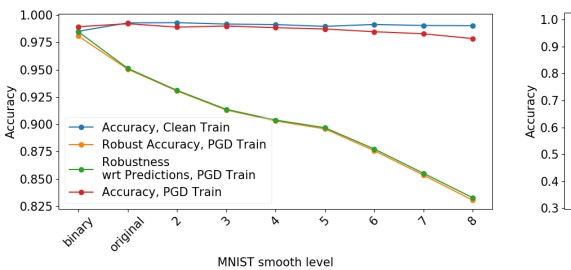

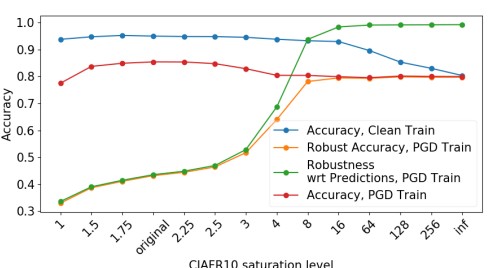

(a) MNIST results under different smooth levels

(b) CIFAR10 results under different saturation levels

Figure 2: Accuracy, Robust Accuracy and Robustness w.r.t. Predictions on different data variants

experiments. Unless otherwise specified, PGD attacks on MNIST variants run with $\epsilon = 0.3$, step size of 0.01 and 40 iterations, and runs with $\epsilon = 8/255$, step size of $2/255$ and 10 iterations on CIFAR10 variants , same as in Madry et al. (2017). We use the PGD attack implementation from the AdverTorch toolbox (Ding et al., 2019).

### 3.3 SENSITIVITY OF ROBUST ACCURACY TO DATA TRANSFORMATIONS

Results on MNIST variants are presented in Figure 2a [6]. The clean accuracy of standard training is very stable across different MNIST variants. This indicates that their classification tasks have similar difficulties, if the training has no robust considerations. When performing PGD adversarial training, clean accuracy drops only slightly. However, both robust accuracy and robustness w.r.t. predictions drop significantly. This indicates that as smooth level goes up, it is significantly harder to achieve robustness. Note that for binarized MNIST with adversarial training, the clean accuracy and the robust accuracy are almost the same. Indicating that getting high robust accuracy on binarized MNIST does not conflict with achieving high clean accuracy. This result conforms with results of provably robust model having high robustness on binarized MNIST described in Section 2.

CIFAR10 result tell a similar story, as reported in Figure 2b [6]. For standard training, the clean accuracy maintains almost at the original level until saturation level 16, despite that it is already perceptually very saturated. In contrast, PGD training has a different trend. Before level 16, the robust accuracy significantly increases from 43.2% until 79.7%, while the clean test accuracy drops only in a comparatively small range, from 85.4% to 80.0%. After level 16, PGD training has almost the same clean accuracy and robust accuracy. However, robustness w.r.t. predictions still keeps increasing, which again indicates the instability of the robustness. On the other hand, if the saturation level is smaller than 2, we get worse robust accuracy after PGD training, e.g. at saturation level 1 the robust accuracy is 33.0%. Simultaneously, the clean accuracy maintains almost the same.

Note that after saturation level 64 the standard training accuracies starts to drop significantly. This is likely due to that high degree of saturation has caused "information loss" of the images. Models trained on highly saturated CIFAR10 are quite robust and the gap between robust accuracy and robustness w.r.t. predictions is due to lower clean accuracy. In contrast, In MNIST variants, the

---

[6] Exact numbers are listed in Table 2 and 3 in Appendix C.

robustness w.r.t. predictions is always almost the same as robust accuracy, indicating that drops in robust accuracy is due to adversarial vulnerability.

From these results, we can conclude that robust accuracy under PGD training is much more sensitive than clean accuracy under standard training to the differences in input data distribution. More importantly, a semantically-lossless shift on the data transformation, while not introducing any unexpected risk for the clean accuracy of standard training, can lead to large variations in robust accuracy. Such previously unnoticed sensitivity raised serious concerns in practice, as discussed in the next section.

## 4 PRACTICAL IMPLICATIONS

Given adversarial robustness' sensitivity to input distribution, we further demonstrate two practical implications: 1) Robust accuracy could be sensitive to image acquisition condition and preprocessing. This leads to unreliable benchmarks in practice; 2) When introducing new dataset for benchmarking adversarial robustness, we need to carefully choose datasets with the right characteristics.

### 4.1 ROBUST ACCURACY IS SENSITIVE TO GAMMA CORRECTION

The natural images are acquired under different lighting conditions, with different cameras and different camera settings. They are usually preprocessed in different ways. All these factors could lead to mild shifts on the input distribution. Therefore, we might get very different performance measures when performing adversarial training on images taken under different conditions. In this section, we demonstrate this phenomenon on variants of CIFAR10 images under different gamma mappings. These variants are then used to represent image dataset acquired under different conditions. Gamma mapping is a simple element-wise operation that takes the original image $x$, and output the gamma mapped image $\tilde{x}^{(\gamma)}$ by performing $\tilde{x}^{(\gamma)} = x^{\gamma}$. Gamma mapping is commonly used to adjust the exposure of an images. We refer the readers to Szeliski (2010) on more details about gamma mappings. Figure 3a shows variants of the same image processed with different gamma values. Lower gamma value leads to brighter images and higher gamma values gives darker images, since pixel values range from 0 to 1. Despite the changes in brightness, the semantic information is preserved.

We perform the same experiments as in the saturated CIFAR10 variants experiment in Section 3. The results are displayed in Figure 3a. Accuracies on clean data almost remain the same across different gamma values. However, under PGD training, both accuracy and robust accuracy varies largely following different gamma values.

These results should raise practitioners' attention on how to interpret robustness benchmark "values". For the same adversarial training setting, the robustness measure might change drastically between image datasets with different "exposures". In other words, if a training algorithm achieves good robustness on one image dataset, it doesn't necessarily achieve similar robustness on another semantically-identical but slightly varied datasets. Therefore, the actual robustness could either be significantly underestimated or overestimated.

This raises the questions on whether we are evaluating image classifier robustness in a reliable way, and how we choose benchmark settings that can match the real robustness requirements in practice. This is an important open question and we defer it to future research.

### 4.2 CHOICE OF DATASETS FOR EVALUATING ROBUSTNESS

As discussed, evaluating robustness on a suitable dataset is important. Here we use fashion-MNIST (fMNIST) (Xiao et al., 2017) and edge-fashion-MNIST (efMNIST) as examples to analyze characteristics of "harder" datasets. The edge-fashion MNIST is generated by running Canny edge detector (Canny, 1986) with $\sigma = 1$ on the fashion MNIST images. Figure 3b shows examples of fMNIST and efMNIST. We performed the same standard training and PGD training experiments on both fMNIST and efMNIST as we did on MNIST. Figure 3b shows the results. We can see that fMNIST exhibit similar behavior to CIFAR10, where the test accuracy is significantly affected by PGD training and the gap between robust accuracy and accuracy is large. On the other hand, efMNIST is closer to the binarized MNIST: the accuracy is affected very little by PGD training, along with an insignificant difference between robust accuracy and accuracy.

Both fMNIST and efMNIST can be seen as a "harder" MNIST, but they are harder in different ways. One one hand, since efMNIST results from the edge detection run on fMNIST, it contains less information. It is therefore harder to achieve higher accuracy on efMNIST than on fMNIST, where richer semantics is accessible. However, fMNIST's richer semantics makes it better resembles natural images' pixel value distribution, which could lead to increased difficulty in achieving

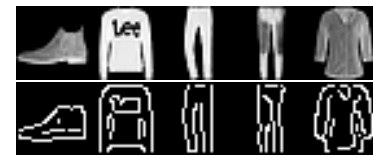

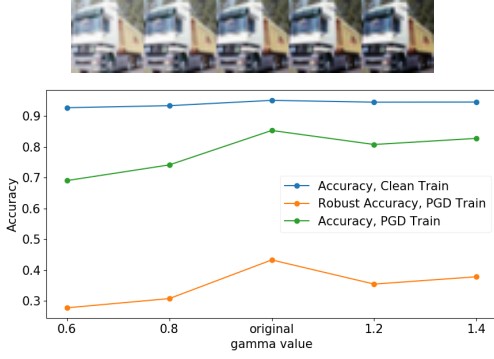

fMNIST: accuracy, standard training, 92.7%
accuracy, PGD training, 81.2%
robust accuracy, PGD training, 65.3%
efMNIST: accuracy, standard training, 88.3%
accuracy, PGD training, 87.2%
robust accuracy, PGD training, 86.6%

(a) Top: Gamma mapped images from left to right 0.6, 0.8, 1.0 (original image), 1.2 , 1.4; Bottom: Robustness results on gamma mapped CIFAR10 variant

(b) Top: Examples of fashion-MNIST images and edge-fashion-MNIST; bottom: Robustness results on fMNIST and efMNIST

Figure 3: Illustrations on Practical Implications

adversarial robustness. efMNIST, on the other hand, can be viewed as a set of "more complex binary symbols" compared to MNIST or binarized MNIST. It is harder to classify these more complex symbols. However, it is easy to achieve high robustness due to the binary pixel value distribution.

To sum up, when introducing new dataset for adversarial robustness, we should not only look for a "harder" one, but we also need to consider whether the dataset is "harder in the right way".

## 5 ATTEMPTS TO UNDERSTAND THE PHENOMENON

In this section, we make initial attempts to understand the sensitivity of adversarial robustness. We use CIFAR10 variants as the running example, but these analyses apply to MNIST variants as well. Saturation pushes pixel values towards 0 or 1, i.e. towards the corner of unit cube, which naturally suggests two potential factors for the change in robustness. 1) the "perturbable volume" decreases; 2) distances between data examples increases. Intuitively, both could be related to the increasd robustness. We analyze them and show that although they are correlated with robustness change, none of them can fully explain the observed phenomena. We then further examine the possibility of increasing robust accuracy on less robust datasets by having larger models and more data.

### 5.1 ON THE INFLUENCE OF PERTURBABLE VOLUME

Saturation moves the pixel values towards 0 and 1, therefore pushing the data points to the corners of the unit cube input domain. This makes the valid perturbation space to be smaller, since the space of perturbation is the intersection between the $\epsilon$-$\ell_\infty$ ball and the input domain. Due to high dimensionality, the volume of "perturbable region" changes drastically across different saturation levels. For example, the average log perturbable volume [7] of original CIFAR10 images are -12354, and the average log perturbable volume of $\infty$-saturated CIFAR10 is -15342, which means that the perturbable volume differs by a factor of $2^{2990} = 2^{(-12352-(-15342))}$. If the differences in perturbable volume is a key factor on the robustness' sensitivity, then by allowing the attack to go beyond the domain boundary [8], the robust accuracies across different saturation levels should behave similarly again, or at least significantly differ from the case of box constrained attacks. We performed PGD attack allowing the perturbation to be outside of the data domain boundary, and compare the robust accuracy to what we get for normal PGD attack within domain boundary. We found that the expected difference is not observed, which serves as evidence that differences in perturbable volume are not causing the differences in robustness on the tested MNIST and CIFAR10 variants.

### 5.2 ON THE INFLUENCE OF INTER-CLASS DISTANCE

When saturation pushes data points towards data domain boundaries, the distances between data points increase too. Therefore, the margin, the distance from data point to the decision boundary,

---

[7]Definition of "log perturbable volume" and other detailed analysis of perturbable volume are given in Appendix D.1 and Table 7.

[8]So we have a controlled and constant perturbable volume across all cases, where the volume is that of the $\epsilon$-$\ell_\infty$ ball

Table 1: Different robust accuracies on datasets with same inter-class distances

| INTER-CLASS DISTANCES | SMOOTH LEVEL OF SMOOTHED MNIST | RESILIENCE OF SMOOTHED MNIST | SCALE FACTOR OF SCALED ORIGINAL MNIST | RESILIENCE OF SCALED ORIGINAL MNIST | SCALE FACTOR OF SCALED BINARIZED MNIST | RESILIENCE OF SCALED BINARIZED MNIST |
|---|---|---|---|---|---|---|
| 7.12 | 3 | 91.3 % | 0.970 | 94.6 % | 0.821 | 98.6 % |
| 7.01 | 4 | 90.3 % | 0.955 | 95.5 % | 0.809 | 98.6 % |
| 6.85 | 5 | 89.6 % | 0.932 | 94.9 % | 0.790 | 98.5 % |

could also increase. We use the "inter-class distance" as an approximation. Inter-class distance [9] characterizes the distances between each class to rest of classes in each dataset. Intuitively, if the distances between classes are larger, then it should be easier to achieve robustness. We also observed (in Appendix D.2.1 Figure 5) that inter-class distances are positively correlated with robust accuracy. However, we also find counter examples where datasets having the same inter-class distance exhibit different robust accuracies. Specifically, We construct scaled variants of original MNIST and binarized MNIST, such that their inter-class distances are the same as smooth-3, smooth-4, smooth-5 MNIST. The scaling operation is defined as $\tilde{x}^{(\alpha)} = \alpha(x - 0.5) + 0.5$, where $\alpha$ is the scaling coefficient. When $\alpha < 1$. each dimension of $x$ is pushed towards the center with the same rate. Table 1 shows the results. We can see that although having the same interclass distances, the smoothed MNIST is still less robust than the their correspondents of scaled binarized MNIST and original MNIST. This indicates the complexity of the problem, such that a simple measure like inter-class distance cannot fully characterize robustness property of datasets, at least on the variants of MNIST.

### 5.3 ON THE REQUIRED MODEL CAPACITY AND SAMPLE COMPLEXITY

In practice, it is unclear how far robust accuracy of PGD trained model is from adversarial Bayes error $RR^*$ for the given data distribution. In the case $RR^*$ is not yet achieved, there is a non-exhaustive list that we can improve upon: 1) use better training/learning algorithms; 2) increase the model capacity; 3) train on more data. Finding a better learning algorithm is beyond the scope of this paper. Here we inspect 2) and 3) to see if it is possible to improve robustness by having larger model and more data. For model capacity, we use differently sized LeNet5 by multiplying the number of channels at each layer with different widen factors. These factors include 0.125, 0.25, 0.5, 1, 2, 4. On CIFAR10 variants, we use WideResNet with widen factors 0.25, 1 and 4. For sample complexity, we follow the practice in Section 3 except that we use a weight decay value of 0.002 to prevent overfitting. For both MNIST and CIFAR10, we test on 1000, 3000, 9000, 27000 and entire training set. Both model capacity and sample complexity results are shown in Figure 4.

For MNIST, both training and test accuracies of clean training are invariant to model sizes, even we only use a model with widen factor 0.125. In slight contrast, both the training and test accuracy of PGD training increase as the model capacity increases, but it plateaus after widen factor 1 at an almost 100% accuracy. For robust accuracy, training robust accuracy kept increasing as model gets larger until the value is close to 100%. However, test robust accuracy stops increasing after widen factor 1, additional model capacity leads to larger (robust) generalization gap. When we vary the size of training set, the model can always fit the training set well to almost 100% clean training accuracy under standard training. The clean test accuracy grows as the training set size get larger. Training set size has more significant impact on robust accuracies of PGD trained models. For most MNIST variants except for binarized MNIST, training robust accuracy gradually drops, and test robust accuracy gradually increases as the training set size increases. This shows that when training set size is small, PGD training overfits to the training set. As training set gets larger, the generalization gap becomes smaller. Both training and test robust accuracies plateau after training set size reaches 27000. Indicating that increasing the training set size might not help in this setting. In conclusion, for MNIST variants, increasing training set size and model capacity does not seem to help beyond a certain point. Therefore, it is not obvious on how to improve robustness on MNIST variants with higher smoothing levels.

CIFAR10 variants exhibit similar trends in general. One notable difference is that for PGD training, the training robust accuracy does not plateau as model size increases. However the test robust accu-

---

[9]The calculation of "inter-class distance" and other detailed analyses are delayed to Appendix D.2.1 and Fig 5. Also note that our inter-class distance is similar to the "distinguishability" in Fawzi et al. (2015), which also measures the distance between classes to quantify easiness of achieving robustness on a certain dataset.

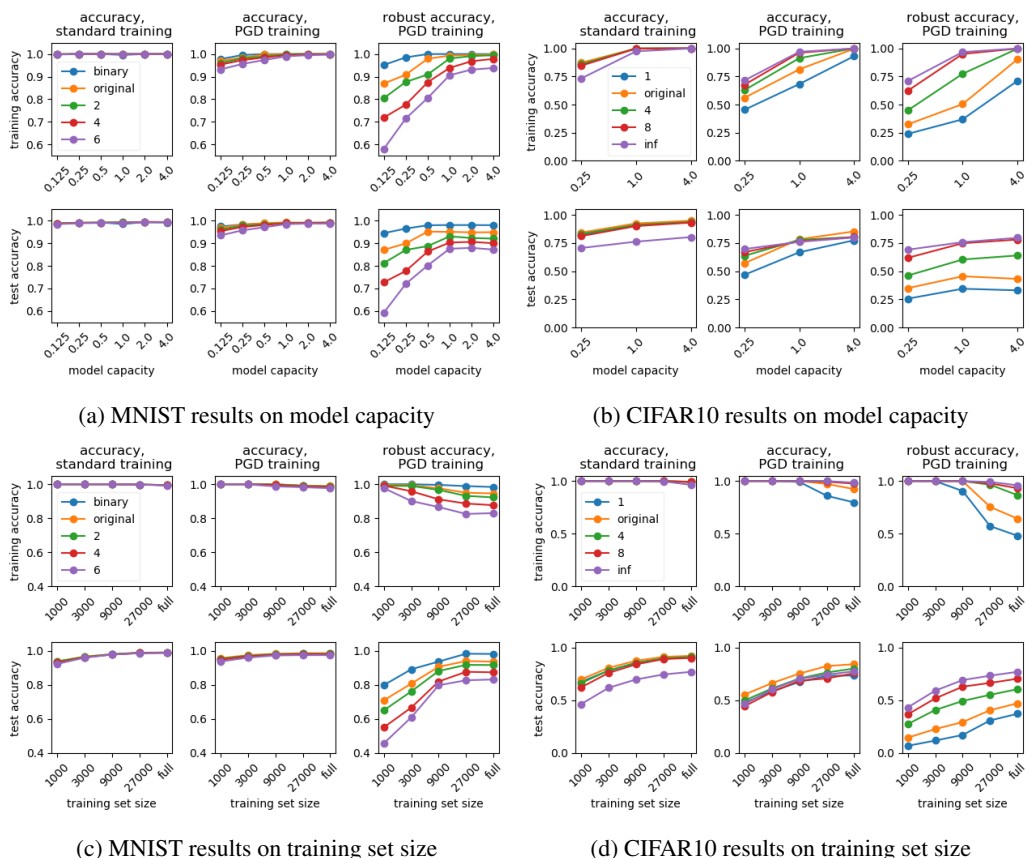

(a) MNIST results on model capacity        (b) CIFAR10 results on model capacity

(c) MNIST results on training set size       (d) CIFAR10 results on training set size

Figure 4: Model capacity and training set size's influences on accuracy and robust accuracy. In each subfigure, the top row contains accuracy and robust accuracy measured on training set, the bottom row contains results measured on test set.

racy plateaus after widen factor 1. Also when training set size increases, the training robust accuracy drops and test robust accuracy increases with no plateau present. These together suggest that having more training data and training a larger model could potentially improve the robust accuracies on CIFAR10 variants. One interesting phenomenon is that binarized MNIST and $\infty$-saturated CIFAR10 has different sample complexity property, despite both being "cornered" datasets. This indicates that the although binarization can largely influence robustness, it does not decide every aspect of it, such as sample complexity. This complex interaction between the classification task and input data distribution is still to be understood further.

## 6 CONCLUSION

In this paper we provided theoretical analyses to show the significance of input data distribution in adversarial robustness, which further motivated our systematic experiments on MNIST and CIFAR10 variants. We discovered that, counter-intuitively, robustness of adversarial trained models are sensitive to semantically-preserving transformations on data. We demonstrated the practical implications of our finding that the existence of such sensitivity questions the reliability in evaluating robust learning algorithms on particular datasets. Finally, we made initial attempts to understand this sensitivity.

**Acknowledgement** We thank Marcus Brubaker for many helpful discussions. We also thank Junfeng Wen and Avishek (Joey) Bose for useful feedbacks on early drafts of the paper.

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

# Appendix

## A PROOFS

### A.1 PROOF FOR EXAMPLE 1

**Proposition A.1** (Existence of Non-Adversarially Robust Decision Boundary). *Let $x$ be uniformly distributed on $[0,1]^d$ and $y = 1$, for all $x$ such that $x^\top \mathbf{1} > \frac{d}{2}$ and $y = 0$ otherwise. Consider adversarial attack under budget $\|\delta\|_\infty < \epsilon$. Then for zero-one loss $L$:*

$$RR^* = \mathbb{E}_{\mathbb{P}(x,y)} \max_{\|\delta\|_\infty < \epsilon} L(Y; h^*(x + \delta)) \geq 1 - \frac{1}{4d\epsilon^2}$$

*Proof.* The argument is well-known in concentration of measure. We provide here for the sake of completeness and adapt it to the context. The hyperplane $HP_2 = \{x \in [0,1]^d : X^\top \mathbf{1} = \frac{d}{2}\}$ defines the decision boundary. We first compute the orthogonal distance of a given point $y = (y_1, y_2, \cdots, y_d) = (x_1 + \delta_1, x_2 + \delta_2, \cdots, x_d + \delta_d)$ to $HP_2$. The point $y$ is the perturbed point within budget $\|\delta\|_\infty < \epsilon$. The vector $\mathbf{1}$ is orthogonal to $HP_2$. Pick any point $x \in HP_2$, the orthogonal distance from $y$ to $HP_2$ is:

$$\ell_2(y, HP_2) = \|Proj_{\mathbf{1}}(y - x)\| = \|\mathbf{1} \frac{(y-x)^\top \mathbf{1}}{\mathbf{1}^\top \mathbf{1}}\|$$

$$= |\frac{y^\top \mathbf{1} - x^\top \mathbf{1}}{\|\mathbf{1}\|}|$$

$$= |\frac{\delta^\top \mathbf{1}}{\|\mathbf{1}\|}|$$

$$= |\frac{y^\top \mathbf{1} - \frac{d}{2}}{\sqrt{d}}|$$

The last two equations show that the $\ell_2$ distance under an $\ell_\infty$ attack can grow at the rate of $\epsilon\sqrt{d}$, for this particular hyperplane $HP_2$.

Now we take the expectation over $[0,1]^d$, and note that expectation of the uniform distribution over a product space $[0,1]^d$ is the same as taking expectation on each dimension (Fubini's theorem), picking each random variable coordinatewise uniformly from $[0,1]$.

$$\mathbb{E}[\ell_2^2(y, HP_2)] = \mathbb{E}[(\frac{y^\top \mathbf{1} - \frac{d}{2}}{d})^2] = \frac{1}{d}\mathbb{E}[(\sum_{i=1}^d y_i - \frac{d}{2})^2]$$

$$= \frac{1}{d}\mathbb{V}[\sum_{i=1}^d y_i] = \frac{1}{d}\sum_{i=1}^d \mathbb{V}[y_i] = \frac{1}{4}$$

Then we apply Markov's inequality, for all real number $t > 0$:

$$\mathbb{P}(\ell_2(x, H) \geq \sqrt{t}) = \mathbb{P}(\ell_2(x, H)^2 \geq t) \leq \frac{1}{4t}$$

Finally, we observe that the longest (in terms of $\ell_2$ norm) such $\epsilon$ $\ell_\infty$ attacks vector to $HP_2$ are parallel to the normal vector $\mathbf{1}$ to $HP_2$. They have $\ell_2$ distance $\epsilon\sqrt{d}$. The set these attacks cover is characterized by $\{x \in [0,1]^d : \ell_\infty(x, H) \leq \epsilon\} = \{x \in [0,1]^d : \ell_2(x, H) \leq \epsilon\sqrt{d}\}$.

Let $t = \epsilon^2 d$, we have:

$$\mathbb{P}(\ell_2(x, H) \geq \sqrt{t}) = \mathbb{P}(\ell_2(x, H)^2 \geq t) \leq \frac{1}{4t} = \frac{1}{4\epsilon^2 d}$$

In the case of zero-one loss, $RR^* = \mathbb{P}(\ell_2(x, H) \leq \epsilon\sqrt{d}) \geq 1 - \frac{1}{4\epsilon^2 d}$. $\qquad\square$

## A.2 PROOF FOR THEOREM 2.1

*Proof.* **(First Inequality for Cube)** The proof here follows that of Ledoux (2005), but we track of the tight constants so as to give tighter adversarial robustness calculations.

Let $\Phi$ be one dimensional standard normal cumulative distribution function and let $\mu_d$ denote $d$ dimensional Gaussian measures. Consider the map $T : \mathbb{R}^d \longrightarrow (0,1)^d$:

$$T(x_1, \cdots, x_d) = (\Phi(x_1), \cdots, \Phi(x_d))$$

$T$ pushes forward $\mu_d$ defined on $\mathbb{R}^d$ into a probability measure $\mathbb{P}$ on $(0,1)^d$:

$$\mathbb{P}(A) = \mu_d(T^{-1}(A))$$

for $A \subset (0,1)^d$. Next we have the following Gaussian isoperimetric/concentration inequality (Ledoux, 2005):

$$\mu_d(B_\epsilon) \geq \Phi(\Phi^{-1}(\mu_d(B)) + \epsilon)$$

for all $B \subset \mathbb{R}^d$ measureable.

Now for $A \subset (0,1)^d$, we have:

$$\mathbb{P}(A_\epsilon) = \mu_d(T^{-1}(A_\epsilon)) \geq \mu_d(T^{-1}(A)_{\epsilon\sqrt{2\pi}}) \geq \Phi(\Phi^{-1}(\mu_d(T^{-1}(A)) + \sqrt{2\pi}\epsilon))$$

where the first inequality follows from that $T$ has Lipschitz constant $\frac{1}{\sqrt{2\pi}}$, and thus $T^{-1}$ has Lipschitz constant $\sqrt{2\pi}$; and the second one follows from Gaussian isoperimetric inequality.

When $\mathbb{P}(A) \geq 1/2$,

$$\Phi(\Phi^{-1}(\mu_d(T^{-1}(A)) + \sqrt{2\pi}\epsilon)) \geq \Phi(\Phi^{-1}(\sqrt{2\pi}\epsilon))$$

Additionally, the inequality $\Phi(x) \geq 1 - e^{\frac{x^2}{2}}$ implies the last inequality in the theorem.

**(Second Inequality for Ball)**

We first define the notion of modulus of convexity for a normed space, in this case $\ell_2$:

$$\delta_{\ell_2}(\epsilon) = \inf\{1 - \|\frac{x+y}{2}\| : \|x\| = \|y\| = 1, \|x - y\| \geq \epsilon\}$$
$$= 1 - \sqrt{1 - \frac{\epsilon^2}{4}}$$

The important property about $\delta_{\ell_2}(\epsilon)$ is that there is a constant $C$ such that:

$$\delta_{\ell_2}(\epsilon) \geq C\epsilon^2$$

By elementary algebraic calculation, We can take $C = \frac{2 - \sqrt{3}}{3}$.

By Equation (2.25) in (Ledoux, 2005),

$$\mathbb{P}(A_\epsilon) \geq 1 - \frac{1}{\mathbb{P}(B)}(1 - \delta_{\ell_2}(\epsilon))^{2d} \geq 1 - \frac{1}{\mathbb{P}(A)}e^{-2d\delta_{\ell_2}(\epsilon)} = 1 - \frac{1}{\mathbb{P}(A)}e^{-2d(\frac{2-\sqrt{3}}{3})\epsilon^2}$$

$\square$

# B DETAILED SETTINGS FOR TRAINING

## B.1 DETAILED SETTINGS OF ADVERSARIAL TRAINING

The LeNet5 (widen factor 1) is composed of 32-channel conv filter + ReLU + size 2 max pooling + 64-channel conv filter + ReLU + size 2 max pooling + fc layer with 1024 units + ReLU + fc layer with 10 output classes. We do not preprocess MNIST images before feeding into the model.

For training LeNet5 on MNIST variants, we use the Adam optimizer with an initial learning rate of 0.0001 and train for 100000 steps with batch size 50.

Table 2: Performance and Robustness of models trained on MNIST variants.

| | STANDARD TRAINING | | PGD TRAINING | | |
| --- | --- | --- | --- | --- | --- |
| MNIST VARIANTS | TEST ACC | | TEST ACC | ROBUST ACCURACY $\epsilon = 0.3$ | ROBUSTNESS W.R.T. PREDICTIONS $\epsilon = 0.3$ |
| BINARIZED | 98.5 % | | 98.9 % | 98.1 % | 98.5 % |
| ORIGINAL | 99.3 % | | 99.2 % | 95.1 % | 95.1 % |
| SMOOTH 2 | 99.3 % | | 98.9 % | 93.0 % | 93.1 % |
| SMOOTH 3 | 99.2 % | | 99.0 % | 91.3 % | 91.4 % |
| SMOOTH 4 | 99.1 % | | 98.8 % | 90.3 % | 90.4 % |
| SMOOTH 5 | 99.0 % | | 98.7 % | 89.6 % | 89.7 % |
| SMOOTH 6 | 99.1 % | | 98.5 % | 87.6 % | 87.7 % |
| SMOOTH 7 | 99.0 % | | 98.3 % | 85.4 % | 85.5 % |
| SMOOTH 8 | 99.0 % | | 97.9 % | 83.1 % | 83.3 % |

Table 3: Performance and Robustness of models trained on CIFAR10 variants.

| | STANDARD TRAINING | | PGD TRAINING | | |
| --- | --- | --- | --- | --- | --- |
| CIFAR10 VARIANTS | TEST ACC | | TEST ACC | ROBUST ACCURACY $\epsilon = 8/255$ | ROBUSTNESS W.R.T. PREDICTIONS $\epsilon = 8/255$ |
| SATURATE 1 | 93.8 % | | 77.5 % | 33.0 % | 33.6 % |
| SATURATE 1.5 | 94.7 % | | 83.7 % | 38.7 % | 39.1 % |
| SATURATE 1.75 | 95.2 % | | 84.9 % | 41.1 % | 41.5 % |
| ORIGINAL | 95.0 % | | 85.4 % | 43.2 % | 43.6 % |
| SATURATE 2.25 | 94.8 % | | 85.4 % | 44.4 % | 44.9 % |
| SATURATE 2.5 | 94.8 % | | 84.8 % | 46.4 % | 47.0 % |
| SATURATE 3 | 94.5 % | | 82.9 % | 51.7 % | 52.9 % |
| SATURATE 4 | 93.8 % | | 80.4 % | 64.0 % | 68.7 % |
| SATURATE 8 | 93.3 % | | 80.4 % | 78.1 % | 93.8 % |
| SATURATE 16 | 92.9 % | | 79.9 % | 79.4 % | 98.4 % |
| SATURATE 64 | 89.6 % | | 79.5 % | 79.3 % | 99.1 % |
| SATURATE 128 | 85.3 % | | 80.2 % | 79.9 % | 99.1 % |
| SATURATE 256 | 83.0 % | | 80.0 % | 79.7 % | 99.2 % |
| SATURATE INF | 80.3 % | | 80.0 % | 79.7 % | 99.2 % |

We use the WideResNet-28-4 as described in Zagoruyko and Komodakis (2016) for our experiments, where 28 is the depth and 4 is the widen factor. We use "per image standardization" [10] to preprocess CIFAR10 images, following Madry et al. (2017).

For training WideResNet on CIFAR10 variants, we use stochastic gradient descent with momentum 0.9 and weight decay 0.0002. We train 80000 steps in total with batch size 128. The learning rate is set to 0.1 at step 0, 0.01 at step 40000, and 0.001 at step 60000.

We performed manual hyperparameter search for our initial experiment and do not observe improvements over the above settings. Therefore we used these settings throughout the all the experiments in the paper unless otherwise indicated.

### B.2 LP ROBUST MODEL DESCRIBED IN SECTION 2

For the linear programming based provably robust model (Wong and Kolter, 2018) (LP-robust model). We trained a ConvNet identical to the one in the original paper. It has 2 convolutional layers, with 16 and 32 channels, each with a stride of 2; and 2 fully connected layers, the first one maps the flattened convolution features to hidden dimension 100, the second maps to 10 logit units. We use ReLUs as the nonlinear activation and there is no max pooling in the network.

We train for 100 epochs with batch size 50. The first 50 epochs are warm start epochs where epsilon increases from 0.01 to 0.3 linearly. We use Adam optimizer (Kingma and Ba, 2014) with a constant learning rate of 0.001.

## C DETAILED EXPERIMENTAL RESULTS

We listed exact numbers of experiments involved in the main body in Table 2, 3, 4 and 5.

---

[10] https://www.tensorflow.org/api_docs/python/tf/image/per_image_standardization

Table 4: Performance and robustness of different sized LeNet5 models on MNIST variants

| | STANDARD TRAINING, ACCURACY | | | | | | | | | | | |
| | TRAINING SET | | | | | | TEST SET | | | | | |
| WIDEN FACTOR | 0.125 | 0.25 | 0.5 | 1 | 2 | 4 | 0.125 | 0.25 | 0.5 | 1 | 2 | 4 |
|---|---|---|---|---|---|---|---|---|---|---|---|---|
| BINARIZED | 99.9% | 100.0% | 100.0% | 99.6% | 100.0% | 100.0% | 98.7% | 99.0% | 99.2% | 98.5% | 99.4% | 99.2% |
| ORIGINAL | 100.0% | 100.0% | 100.0% | 100.0% | 100.0% | 100.0% | 98.8% | 99.2% | 99.2% | 99.3% | 99.4% | 99.3% |
| SMOOTH 2 | 99.9% | 100.0% | 100.0% | 100.0% | 100.0% | 100.0% | 98.8% | 99.0% | 99.1% | 99.3% | 99.3% | 99.4% |
| SMOOTH 3 | 99.9% | 99.9% | 100.0% | 100.0% | 100.0% | 100.0% | 98.8% | 98.8% | 99.2% | 99.2% | 99.1% | 99.3% |
| SMOOTH 4 | 99.9% | 100.0% | 100.0% | 100.0% | 100.0% | 100.0% | 98.7% | 99.0% | 99.0% | 99.1% | 99.4% | 99.4% |
| SMOOTH 5 | 99.8% | 100.0% | 100.0% | 100.0% | 100.0% | 100.0% | 98.5% | 99.0% | 99.2% | 99.0% | 99.3% | 99.3% |
| SMOOTH 6 | 99.8% | 100.0% | 100.0% | 100.0% | 100.0% | 100.0% | 98.4% | 98.9% | 99.0% | 99.1% | 99.2% | 99.3% |
| SMOOTH 7 | 99.8% | 99.9% | 100.0% | 100.0% | 100.0% | 100.0% | 98.5% | 98.8% | 99.0% | 99.0% | 99.3% | 99.3% |
| SMOOTH 8 | 99.7% | 100.0% | 100.0% | 100.0% | 100.0% | 100.0% | 98.4% | 98.9% | 98.9% | 99.0% | 99.2% | 99.0% |

| | PGD TRAINING, ACCURACY | | | | | | | | | | | |
| | TRAINING SET | | | | | | TEST SET | | | | | |
| WIDEN FACTOR | 0.125 | 0.25 | 0.5 | 1 | 2 | 4 | 0.125 | 0.25 | 0.5 | 1 | 2 | 4 |
|---|---|---|---|---|---|---|---|---|---|---|---|---|
| BINARIZED | 97.8% | 99.6% | 100.0% | 100.0% | 100.0% | 100.0% | 97.4% | 98.3% | 98.8% | 98.9% | 99.0% | 99.2% |
| ORIGINAL | 97.0% | 98.4% | 99.8% | 100.0% | 100.0% | 100.0% | 97.0% | 98.2% | 98.9% | 99.2% | 99.1% | 99.2% |
| SMOOTH 2 | 96.1% | 98.1% | 99.0% | 99.9% | 100.0% | 100.0% | 96.1% | 97.8% | 98.5% | 98.9% | 99.0% | 99.0% |
| SMOOTH 3 | 96.3% | 97.8% | 98.9% | 99.7% | 99.9% | 100.0% | 96.5% | 97.6% | 98.6% | 99.0% | 99.1% | 99.1% |
| SMOOTH 4 | 95.3% | 97.3% | 98.5% | 99.5% | 99.8% | 99.9% | 95.4% | 97.2% | 98.1% | 98.8% | 99.0% | 99.0% |
| SMOOTH 5 | 94.9% | 96.5% | 98.0% | 99.3% | 99.6% | 99.8% | 95.0% | 96.5% | 97.9% | 98.7% | 98.9% | 98.9% |
| SMOOTH 6 | 93.2% | 95.6% | 97.4% | 99.0% | 99.5% | 99.7% | 93.5% | 95.7% | 97.1% | 98.5% | 98.7% | 98.7% |
| SMOOTH 7 | 91.9% | 95.0% | 97.5% | 98.7% | 99.2% | 99.4% | 92.4% | 95.2% | 97.2% | 98.3% | 98.5% | 98.7% |
| SMOOTH 8 | 89.4% | 94.2% | 96.5% | 98.4% | 99.0% | 99.3% | 89.7% | 94.4% | 96.4% | 97.9% | 98.2% | 98.4% |

| | PGD TRAINING, ROBUST ACCURACY | | | | | | | | | | | |
| | TRAINING SET | | | | | | TEST SET | | | | | |
| WIDEN FACTOR | 0.125 | 0.25 | 0.5 | 1 | 2 | 4 | 0.125 | 0.25 | 0.5 | 1 | 2 | 4 |
|---|---|---|---|---|---|---|---|---|---|---|---|---|
| BINARIZED | 95.2% | 98.5% | 100.0% | 100.0% | 100.0% | 100.0% | 94.5% | 96.5% | 98.0% | 98.1% | 98.0% | 98.0% |
| ORIGINAL | 86.9% | 90.8% | 97.9% | 99.3% | 99.6% | 99.8% | 87.1% | 89.9% | 95.2% | 95.1% | 94.8% | 94.9% |
| SMOOTH 2 | 80.5% | 87.6% | 90.9% | 98.0% | 99.1% | 99.5% | 81.2% | 87.0% | 88.7% | 93.0% | 92.3% | 92.1% |
| SMOOTH 3 | 75.2% | 82.0% | 90.3% | 95.5% | 97.8% | 98.7% | 75.7% | 81.5% | 88.5% | 91.3% | 91.6% | 90.8% |
| SMOOTH 4 | 71.9% | 77.6% | 87.5% | 93.9% | 96.8% | 97.9% | 72.7% | 77.7% | 86.3% | 90.3% | 90.6% | 90.0% |
| SMOOTH 5 | 65.7% | 77.1% | 85.7% | 92.5% | 94.6% | 95.0% | 66.2% | 77.1% | 85.1% | 89.6% | 89.8% | 88.4% |
| SMOOTH 6 | 58.0% | 71.5% | 80.5% | 90.6% | 93.1% | 93.8% | 59.3% | 72.0% | 80.2% | 87.6% | 88.0% | 87.2% |
| SMOOTH 7 | 61.7% | 74.2% | 83.3% | 87.6% | 90.5% | 92.6% | 62.8% | 75.3% | 83.0% | 85.4% | 86.7% | 87.8% |
| SMOOTH 8 | 70.3% | 72.4% | 80.3% | 85.3% | 90.5% | 88.7% | 71.7% | 73.2% | 80.3% | 83.1% | 86.9% | 83.8% |

# D    DETAILED ANALYSES

## D.1    DETAILED ANALYSIS OF EFFECTS OF DATA DOMAIN BOUNDARY

One natural hypothesis about the reason of achieving better robustness could be that it is the effect of the boundaries. Indeed, if the data distribution is closer to the data domain boundary, the valid perturbation space, the $\epsilon$-$\ell_\infty$ ball may be restricted since it will intersect with the boundary. We then test the correlation between "how close the data distribution is to the boundary" and its achievable robustness, by examining the volume of the allowed perturbed box across different datasets.

The intersection of the data domain, unit cube $[0, 1]^d$, with the allowed perturbation space, $\epsilon$-$\ell_\infty$ ball $[x_i - \epsilon, x_i + \epsilon]^d$, is the hyperrectangle $[\max\{x_i - \epsilon, 0\}, \min\{x_i + \epsilon, 1\}]^d$, where $i = 1, \cdots, d$ are the indexes over input dimensions. The size of the available perturbation space at $x$ and $\epsilon$ is defined by the volume of this hyperrectangle:

$$\mathrm{Vol}(x, \epsilon) = \prod_{i=1}^{d}(\min\{x_i + \epsilon_i, 1\} - \max\{x_i - \epsilon_i, 0\})$$

In high dimensional space, when $\epsilon$ is fixed, this volume varies greatly based on the location of $x$. For example, if $x$ is on one of the corners of the unit cube, $\mathrm{Vol}(x_{corner}, \epsilon) = \epsilon^d$. If each dimension of $x$ is at least $\epsilon$ away from all the data boundaries, then the volume of the hyperrectangle is $\mathrm{Vol}(x_{inside}, \epsilon) = (2\epsilon)^d$. Therefore there can be $2^d$ times difference of perturbable space between different data points. As shown in the average log perturbable volumes Table 6, we can see that different variations of datasets has significantly different perturbable volumes, with the same trend with previously described. It is notable that for the original CIFAR10 datasets has log volume -12354, which is very close to the -12270. The different of 84 bits indicates on average, the perturbation space is $2^{84}$ smaller than the full $\epsilon$-$\ell_\infty$ ball if there is no intersection with the data domain boundary. Volume differences between different saturation or smooth level can be interpreted in the similar

Table 5: Performance and robustness of different sized Wide ResNet models on CIFAR10 variants

| | STANDARD TRAINING, ACCURACY | | | | | |
| | TRAINING SET | | | TEST SET | | |
| WIDEN FACTOR | 0.25 | 1 | 4 | 0.25 | 1 | 4 |
|---|---|---|---|---|---|---|
| SATURATE 1 | 85.5% | 99.9% | 100.0% | 82.4% | 91.1% | 93.8% |
| SATURATE 1.5 | 87.0% | 99.9% | 100.0% | 84.2% | 92.1% | 94.7% |
| SATURATE 1.75 | 87.4% | 99.9% | 100.0% | 84.5% | 93.0% | 95.2% |
| ORIGINAL | 87.2% | 99.9% | 100.0% | 84.4% | 92.5% | 95.0% |
| SATURATE 2.25 | 87.3% | 99.9% | 100.0% | 84.5% | 92.5% | 94.8% |
| SATURATE 2.5 | 86.4% | 99.9% | 100.0% | 83.7% | 92.3% | 94.8% |
| SATURATE 3 | 86.2% | 99.9% | 100.0% | 84.0% | 92.2% | 94.5% |
| SATURATE 4 | 85.8% | 99.9% | 100.0% | 83.1% | 91.1% | 93.8% |
| SATURATE 8 | 84.6% | 99.8% | 100.0% | 81.2% | 90.1% | 93.3% |
| SATURATE 16 | 83.5% | 99.7% | 100.0% | 81.0% | 89.4% | 92.9% |
| SATURATE 64 | 80.5% | 99.4% | 100.0% | 79.2% | 86.9% | 89.6% |
| SATURATE 128 | 77.1% | 98.7% | 100.0% | 74.6% | 83.0% | 85.3% |
| SATURATE 256 | 73.7% | 97.6% | 100.0% | 70.7% | 76.5% | 83.0% |
| SATURATE INF | 73.2% | 97.3% | 99.9% | 70.6% | 76.3% | 80.3% |

| | PGD TRAINING, ACCURACY | | | | | |
| | TRAINING SET | | | TEST SET | | |
| WIDEN FACTOR | 0.25 | 1 | 4 | 0.25 | 1 | 4 |
|---|---|---|---|---|---|---|
| SATURATE 1 | 45.4% | 68.3% | 93.1% | 46.8% | 66.9% | 77.5% |
| SATURATE 1.5 | 52.1% | 76.5% | 98.0% | 53.3% | 74.1% | 83.7% |
| SATURATE 1.75 | 53.8% | 79.5% | 99.2% | 55.3% | 77.0% | 84.9% |
| ORIGINAL | 56.1% | 81.4% | 99.7% | 57.1% | 78.4% | 85.4% |
| SATURATE 2.25 | 56.8% | 82.7% | 99.9% | 58.1% | 78.8% | 85.4% |
| SATURATE 2.5 | 57.6% | 83.9% | 100.0% | 58.3% | 79.1% | 84.8% |
| SATURATE 3 | 60.0% | 86.3% | 100.0% | 60.8% | 79.5% | 82.9% |
| SATURATE 4 | 62.8% | 91.3% | 100.0% | 63.7% | 77.9% | 80.4% |
| SATURATE 8 | 67.7% | 96.1% | 100.0% | 67.0% | 76.6% | 80.4% |
| SATURATE 16 | 67.2% | 96.1% | 99.9% | 66.0% | 76.4% | 79.9% |
| SATURATE 64 | 70.0% | 96.5% | 99.9% | 68.6% | 75.8% | 79.5% |
| SATURATE 128 | 71.4% | 96.4% | 99.9% | 68.9% | 76.6% | 80.2% |
| SATURATE 256 | 68.6% | 96.9% | 99.9% | 65.7% | 76.6% | 80.0% |
| SATURATE INF | 71.5% | 96.9% | 99.9% | 69.7% | 76.1% | 80.0% |

| | PGD TRAINING, ROBUST ACCURACY | | | | | |
| | TRAINING SET | | | TEST SET | | |
| WIDEN FACTOR | 0.25 | 1 | 4 | 0.25 | 1 | 4 |
|---|---|---|---|---|---|---|
| SATURATE 1 | 24.0% | 36.9% | 71.1% | 25.6% | 34.4% | 33.0% |
| SATURATE 1.5 | 29.0% | 44.4% | 81.3% | 31.6% | 40.7% | 38.7% |
| SATURATE 1.75 | 30.9% | 47.8% | 86.0% | 32.7% | 44.0% | 41.1% |
| ORIGINAL | 32.4% | 50.4% | 90.3% | 35.0% | 45.5% | 43.2% |
| SATURATE 2.25 | 33.9% | 52.9% | 93.4% | 36.1% | 47.3% | 44.4% |
| SATURATE 2.5 | 35.5% | 55.4% | 96.0% | 37.5% | 49.1% | 46.4% |
| SATURATE 3 | 38.4% | 61.5% | 98.9% | 40.6% | 52.5% | 51.7% |
| SATURATE 4 | 44.9% | 77.4% | 99.7% | 46.1% | 60.4% | 64.0% |
| SATURATE 8 | 62.3% | 95.0% | 99.8% | 61.9% | 74.9% | 78.1% |
| SATURATE 16 | 66.0% | 95.5% | 99.9% | 65.0% | 75.5% | 79.4% |
| SATURATE 64 | 69.1% | 96.3% | 99.9% | 67.6% | 75.5% | 79.3% |
| SATURATE 128 | 70.7% | 96.2% | 99.9% | 68.2% | 76.2% | 79.9% |
| SATURATE 256 | 68.0% | 96.7% | 99.9% | 65.2% | 76.3% | 79.7% |
| SATURATE INF | 70.9% | 96.7% | 99.9% | 69.2% | 75.8% | 79.7% |

Table 6: Perturbable volumes of different variants of MNIST and CIFAR10. Values shown in table are the average log value (in bits) of volumes of test data. For MNIST, $\epsilon = 0.3$, for CIFAR10 $\epsilon = 8/255$.

| MNIST (VALID RANGE -1361 TO -577) | | | | CIFAR10 (VALID RANGE -15342 TO -12270) | | | | | | | |
| BINARY | ORIGINAL | 3 | 5 | ORIGINAL | 4 | 8 | 16 | 64 | 256 | 512 | INF |
|---|---|---|---|---|---|---|---|---|---|---|---|
| -1361 | -1297 | -1265 | -1234 | -12354 | -12394 | -12477 | -12657 | -13620 | -14747 | -15028 | -15342 |

way. Note that for CIFAR10 images with large saturation, although they appear similar to human, they actually have very large differences in terms of perturbable volumes.

If the perturbable volume hypothesis holds, then we should observe significantly lower accuracy under PGD attack if we allow perturbation outside of data domain boundary. Since this greatly increases the perturbable volume. We measure the accuracy under PGD attack with and without considering data domain boundary for both MNIST and CIFAR10 variants. The results are shown in Table 7. "With considering boundary" corresponds to regular PGD attacks. We can see that allowing PGD to perturb out of bound do not reduce accuracy under attack. This means that PGD

Table 7: PGD attack results with and without domain boundary constraints on MNIST and CIFAR10

| MNIST | | | CIFAR10 | | |
|---|---|---|---|---|---|
| MNIST VARIANTS | ROBUST ACCURACY W/ BOUND | ROBUST ACCURACY W/O BOUND | CIFAR10 VARIANTS | ROBUST ACCURACY W/ BOUND | ROBUST ACCURACY W/O BOUND |
| BINARIZED | 98.1 % | 96.1 % | SATURATE 1 | 33.0 % | 32.7 % |
| ORIGINAL | 95.1 % | 95.1 % | ORIGINAL | 43.2 % | 43.0 % |
| SMOOTH 2 | 93.0 % | 92.9 % | SATURATE 4 | 64.0 % | 64.0 % |
| SMOOTH 3 | 91.3 % | 91.5 % | SATURATE 8 | 78.1 % | 78.1 % |
| SMOOTH 4 | 90.3 % | 90.6 % | SATURATE 16 | 79.4 % | 79.4 % |
| SMOOTH 5 | 89.6 % | 89.9 % | SATURATE INF | 79.7 % | 79.4 % |

is not able to use the significantly larger additional volumes even for binarized MNIST or highly saturated CIFAR10, whose data points are on or very close to the corner. In some cases, allowing perturbation outside of domain boundary makes the attack slightly less effective. This might be due to that data domain boundary constrained the perturbation to be in an "easier" region. This might seem surprising considering the huge difference in perturbable volumes, these results conform with empirical results in previous research (Goodfellow et al., 2014; Warde-Farley and Goodfellow, 2016) that adversarial examples appears in certain directions instead of being distributed in small pockets across space. Therefore, the perturbable volume hypothesis is rejected.

### D.2 DETAILED ANALYSES OF INTER-CLASS DISTANCE

#### D.2.1 CALCULATION OF INTER-CLASS DISTANCE

We calculate the inter-class distance as follows. Let $D = \{x_i\}$ denote the set of all the input data points, $D_c = \{x_i|y_i = c\}$ denote the set of all the data points in class $c$, and $D_{\neg c} = \{x_i|y_i \neq c\}$ denote all the data points not in class $c$. Our goal is to calculate $d(D_c, D_{\neg c})$ for all the classes, where $d(D_c, D_{\neg c})$ approximates the margin between class $c$ and the rest. To estimate $d(D_c, D_{\neg c})$, we first compute the margin for each data point $x$ in class $c$. To do that, we calculate the average $\|x - x_j\|_2$, where $x_j \in D_{\neg c}$ is one of $x$'s 10% nearest neighbors in $D_{\neg c}$. Lastly, the inter-class distance of class $c$, $d(D_c, D_{\neg c})$, is then calculated as the average of smallest 10% $d(x, D_{\neg c})$ for $x \in D_c$.

Note that we choose $\ell_2$ distance for inter-class distance, instead of using the $\ell_\infty$ which measures the robustness. This is because $\ell_\infty$-distance between data examples is essentially the max over the per pixel differences, which is always very close to 1. Therefore the $\ell_\infty$-distance between data examples is not really representative / distinguishable.

Figure 5 shows the inter-class distances (averaged over all classes) calculated on MNIST and CIFAR10 variants. The binarized MNIST has a significantly larger inter-class distance. As smoothing kernel size increases, the distance also decrease slightly. On CIFAR10 variants, as the saturation level gets higher, the inter-class distance increases monotonically. We also directly plot inter-class distance vs robust accuracy on MNIST and CIFAR10 variants. In general, inter-class distance shows a strong positive correlation with robust accuracy under these transformations. With one exception that original MNIST has smaller inter-class distance, but is sightly more robust than smooth-2 MNIST. This, together with the counter examples we gave in Table 1, suggests that inter-class distance cannot fully explain the robust variation across different dataset variants.

#### D.2.2 INTER-CLASS DISTANCE COULD POTENTIALLY INFLUENCE REQUIRED MODEL CAPACITY

We attempt to understand the relation between the inter-class distance of a dataset and its achievable robustness in this section. We first illustrate our intuition in a synthetic experiment, where a ReLU network is trained to perfectly separate 2 concentric spheres (Gilmer et al., 2018), as shown in Figure 6. Here the inter-class distance is the width of the ring between two spheres. In such example, adversarial training is actually closely related to the inter-class distance of the data. In fact, in the simple setting where the classifier is linear, it has been shown in Xu et al. (2009) that adversarial training, as a particular form of robust optimization, is equivalent to maximizing the classification margins. Following this intuition, one can easily see that the effect of adversarial training is to push two spheres close to each other, and requires the network to perfectly separate the new spheres with much smaller inter-class.

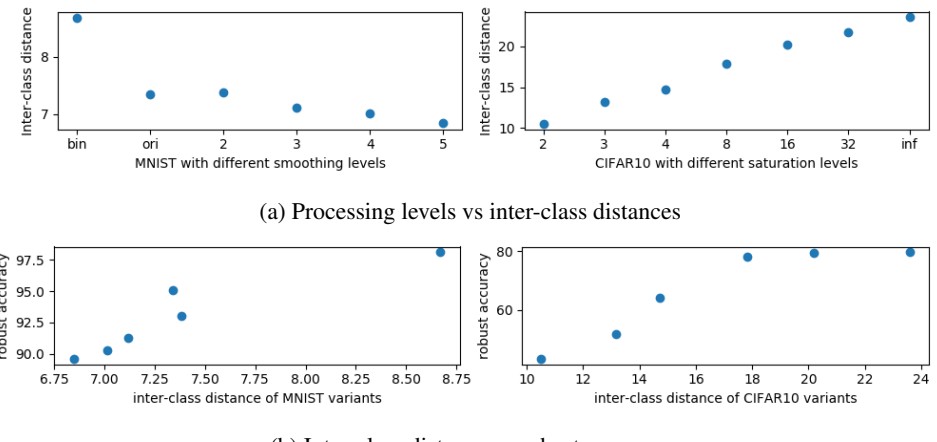

(a) Processing levels vs inter-class distances

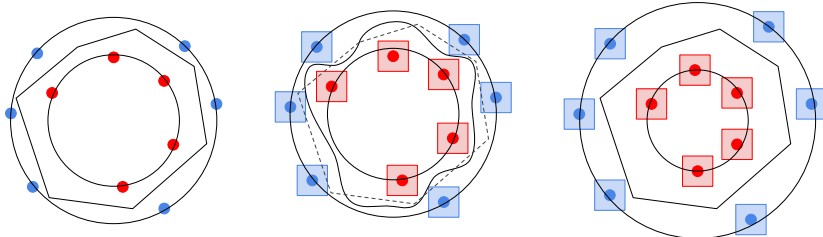

(b) Inter-class distance vs robust accuracy

Figure 5: Inter-class distance's influence on robust accuracy on different MNIST and CIFAR10 variants

Intuitively, when the inter-class distance is large, i.e. the gap between two spheres are large, a reasonable model should be able to achieve good standard accuracy. We have also observed such phenomenon on original MNIST and saturated CIFAR10 (say level 16). As the inter-class distance gets smaller, although the model capacity could still be enough for the standard training, it may no longer be enough for adversarial training, upon which we would observe that although the test accuracies stay similar, accuracies under adversarial attack significantly would drop. We have also seen similar behavior on smooth MNIST data and smaller level of saturated CIFAR10 data. Finally, when the inter-class distance is so small such that even a high clean test accuracy may be difficult to achieve.

Considering robust accuracy as the clean accuracy with a smaller gap between the spheres, the next theorem provides a theoretical guarantee in relating together the difficulty of attaining good accuracy under attack and the model capacity (Ball, 1997), verifying our intuition above. Note that one way to measure the capacity of a ReLU network is by counting the number of its induced piece-wise linear region, which is closely related to the number of facets of its decision boundary.

**Theorem D.1.** *Let $d(K, L)$ between symmetric convex bodies $K$ and $L$ denote the least positive $d$ for which there is a linear image $\tilde{L}$ of $L$ such that $\tilde{L} \subset K \subset d\tilde{L}$. Let $K$ be a (symmetric) polytope in $\mathbb{R}^n$ with $d(K, B_2^n) = d$. Then $K$ has at least $e^{n/(2d^2)}$ facets. On the other hand, for each $n$, there is a polytope with $4n$ facets whose distance from the ball is at most 2.*

Figure 6: Illustration of the relationship between the inter-class distance and the required model capacity. Left: when distance is small, a small capacity polytope classifier could separate original data; middle: when distance is small, the small capacity polytope classifier is not able to separate data points "robustly", but a more complex nonlinear classifier could; right:when distance is large, the small capacity polytope classifier can separate data points "robustly".

The above analysis is partially supported by our experiments on model capacity in Section 5.3. However, as we've shown in Section 5.2, the nature of the problem is complex and more conclusive statements requires further research.

