# OpenReview forum: "On the Sensitivity of Adversarial Robustness to Input Data Distributions"
_ICLR.cc/2019/Conference_

### Official Review · AnonReviewer3 · 2018-11-03
**A super interesting paper discussing the impact of data distribution on adversarial robustness of trained neural networks**

**Rating:** 7
**Confidence:** 2

**Review:**

This paper provides several theoretical and practical insights on the impact of data distribution to adversarial robustness of trained networks. The paper reads well and provides analysis on two datasets MNIST and CIFAR10. I particularly like the result demonstrating that a lossless transformation on the data distribution could significantly impact the robustness of an adversarial trained models. The idea of using smoothness and saturation to bridge the gap between the MNIST and CIFAR10 datasets was also very interesting. One thing that is not clear from the paper is how one could use the findings from this paper and put it into practice. In other words, it would help if the authors could provide some insights on how to improve a model robustness w.r.t the changes in the data distribution. The authors did an attempt toward this in section 5, but that seems to only cover three factors that do not cause the difference in robustness.

---

> ### Author Response · Authors · 2018-11-16
> **Response to AnonReviewer3**
>
> We thank the reviewer for your time spent, interests and kind evaluation on our work.
>
> With regard to your comments on "put findings into practice" and "how to improve a model robustness w.r.t the changes in the data distribution", we believe these are very meaningful future research directions. We covered a few aspects, but other aspects are out of the scope of our current paper. In particular, Section 4 demonstrates the issues of robustness evaluation caused by the sensitivity. Section 5 intends to inspire future research by excluding the possibilities that the sensitivity can be explained by obvious reasons, or be resolved in trivial ways, which implies that understanding the causes or finding the remedies is non-trivial future directions.

---

### Official Review · AnonReviewer1 · 2018-11-06
**investigation as to the origin of lack of robustness of classifiers to perturbations of the input data**

**Rating:** 5
**Confidence:** 4

**Review:**

The paper is interesting and topical: robustness to adversarial input presentation (or shifts in training data itself, even those of the nature described by the authors 'semantic-lossless' shifts). Adversarial inputs are investigated under l-inf bounded perturbations, while multiclass classification on images is the target problem considered. The theoretical parts of the paper, assigning lack of adversarial robustness to the shape of the input distribution (Section 2) is the strongest part of the paper, adding some simple and important insights. Unfortunately, the empirical part of the paper is weakened by an over-reliance of (custom perturbations of ) the popular MNIST and CIFAR10 datasets (which are themselves based on larger sets). Furthermore, the basic conclusion as to causes and remedies of lack of robustness is not evident, and it is not evident that it has been sufficiently investigated. Shape yes, differences in perturbable volume not (how does that concur with Section 2?), and inter-class distance also not. Are we to base these conclusions on 2 perturbed datasets? How are readers to synthesize the final conclusion that robustness is a 'complex interaction of tasks and data', other than what they would already expect? In short, a valiant effort, and a good direction, but one that needs more work.

---

> ### Author Response · Authors · 2018-11-16
> **Response to AnonReviewer1**
>
> Thank you for the effort in reviewing the paper and also finding that the topic is interesting.
>
> --------
> >> "The paper is interesting and topical: robustness to adversarial input presentation (or shifts in training data itself, even those of the nature described by the authors 'semantic-lossless' shifts)"
>
> We don't quite understand the phrase "adversarial input presentation" in the review. Just to avoid possible misunderstanding, let us first clarify our finding in this paper: the sensitivity of adversarial robustness to the input data distribution D. Assume that R is the adversarial robustness of the method of adversarial training on D. Now consider a semantic-lossless shift of D, say D'. Again we train a model on the training data sampled from D', and test its adversarial robustness R' on the test set again sampled from D' too. We find that R' can be significantly different from R. We have improved the introduction to make this point more clear.
>
> So, in our terminology, there is no adversarial input presentation. D' is just a common 'semantic-lossless' shift of D in the paper, without being against adversarial training.
>
>
> --------
> >> "Unfortunately, the empirical part of the paper is weakened by an over-reliance of (custom perturbations of ) the popular MNIST and CIFAR10 datasets (which are themselves based on larger sets). Furthermore, the basic conclusion as to causes and remedies of lack of robustness is not evident, and it is not evident that it has been sufficiently investigated. Shape yes, differences in perturbable volume not (how does that concur with Section 2?), and inter-class distance also not. Are we to base these conclusions on 2 perturbed datasets? "
>
> We have two sets of experiments in this paper, one in Section 3 and one in Section 5. We are not sure which one this review corresponds to.
>
> As mentioned above, the purpose of Section 3 is to demonstrate the existence of the sensitivity. The transformations (semantic-lossless shifts) in this section are common and are not designed to overthrow the adversarial robustness. We believe our experimental results in this section is sufficiently significant to prove the existence of such sensitivity. Furthermore, the gamma correction in Section 4 is a common transformation in image processing, which also demonstrates the sensitivity.
>
> On the other hand, we agree with the reviewer that results in Section 5 are not evident enough to conclude about the causes and remedies. In fact, it is exactly because of this that we only make conservative claims in this section. As mentioned above, Section 5 is an initial attempt, and we do not consider it as the main contribution of this paper. Although we don't have a definitive answer for the problem, we believe that our findings in this paper should be noticed by the adversarial example community and it is already sufficiently significant and important for a publication.
>
> Lastly, let us also clarify our statements on perturbable volume and inter-class distance in Section 5. We found that they are both correlated with robust accuracy. However, when we examine whether they are decisive factors for robustness, counterexamples exist for both of them. We therefore made inconclusive statements highlighting the complexity of the problem. We faithfully report our investigations in this section, and we hope that they can inspire future research around this topic.
>
> In case our responses above do not address your concerns, it would be nice if you can clarify them further.
>
>
> --------
> >> "How are readers to synthesize the final conclusion that robustness is a 'complex interaction of tasks and data', other than what they would already expect?"
>
> We do not intend to make this a "final conclusion". By "complex interaction of tasks and data", we only want to make a remark on the sample complexity difference between binarized MNIST and binarized CIFAR10. Our intention is to truthfully report to the readers that although binarization largely affects robustness, it does not decide every aspect of it. We have updated our paper to avoid these confusions.

---

### Official Review · AnonReviewer2 · 2018-11-06

**Rating:** 7
**Confidence:** 3

**Review:**

A nice paper that clarifies the difference between the clean accuracy (accuracy of models on non-perturbed examples) and the robust accuracy (accuracy of models on adversarially perturbed examples) and it shows that changing the marginal distribution of the input data P(x) while preserving its semantic P(y|x) fixed affects the robustness of the model. Therefore, testing the robustness of the model should be performed in a careful manner. Comprehensive experiments were performed to show that changing the distribution of the MINST (smoothing) and CIFAR (saturation) data could lead to a significant difference in robust accuracy while the clean accuracy is almost steady. In addition, a set of experiments were performed in an attempt to search for the criteria required for choosing a proper dataset for testing adversarial attack to measure the robustness.

Although I’m not expert in the field of adversarial attack but the paper is very nice to read and easy to follow (I have not checked the proof of the theorems though).

---

### Author Response · Authors · 2018-11-16
**Response to all reviewers**

We thank the reviewers for their time and efforts. We especially appreciate that all reviewers find that the problem being investigated is interesting.
We summarize our main contributions to address common concerns in this post, and provide more details in the responses to each reviewer.

To avoid clutter, we use the following abbreviated phrases:
"sensitivity" means "the sensitivity of adversarial robustness to input data distributions".
"clean accuracy" means "prediction accuracy of the standardly trained model on natural examples".
"robust accuracy" means "prediction accuracy of the adversarially trained model on adversarial examples".

Our main contribution is the discovery that the robust accuracy of adversarially trained models is very sensitive to input data distributions, which is previously unnoticed in the literature and in sharp contrast to the steady clean accuracy in the standard learning setting. In theory, we show regular Bayes error's invariance and robust error's sensitivity to input distribution shift. We also show that if the data is uniformly distributed in a unit cube, then the perfect decision boundary cannot be robust. On the other hand, for the binary MNIST dataset, we found a provably robust classifier, using the algorithm by Kolter and Wong (2017), which guarantees 97% robust accuracy under 0.3 \ell_infty perturbation. Such contrast motivates us to design systematic experiments (smoothed MNIST and saturated CIFAR in the paper) to investigate the dependence of adversarial robustness on the input data distributions, which empirically demonstrates the sensitivity.

We admit (also in the paper) that we don't have a definitive explanation or remedy for such sensitivity. Section 5 is only an initial attempt, and we do not consider it as the main contribution of the paper. We examined some natural hypotheses. We found highly correlated factors, but they don't fully explain the phenomenon, which suggests the absence of obvious solutions. We report these results not to make conclusive claims, but to hope to inspire future research in this direction.

We believe that solely by itself our discovery of the sensitivity is a significant contribution, and it has an important implication in both practice and theory. In practice, our finding raises questions on how to properly evaluate the adversarial robustness of different learning algorithms. Benchmarking adversarial robustness on only a few datasets may not be reliable due to such sensitivity. In theory, our finding opens a new angle for understanding the cause of "lack of robustness". More specifically, Schmidt et al. (2018) show that different data distributions could have drastically different properties of adversarially robust generalization. Our finding indicates that gradual semantics-preserving transformations of data distribution can also cause large changes to datasets' achievable robustness. Tsipras et al. (2018) hypothesize the existence of the intrinsic tradeoff between clean accuracy and adversarial robustness. Our work complements this result, by showing different levels of tradeoffs for different input data distributions.

In summary, we would like to emphasize that the main contribution of this paper is discovering and firmly demonstrating the existence of the sensitivity both in theory and in experiments, which we believe solely by itself is important in practice and understanding the phenomenon of adversarial examples. Although we don't have a definitive explanation or remedy for it, our paper is a starting point for future lines of research around this topic.
We have improved the introduction of the paper to make this message more direct. Please see the latest updated version.


Kolter, J. Z. and Wong, E. (2017). Provable defenses against adversarial examples via the convex outer adversarial polytope. arXiv preprint arXiv:1711.00851.

Schmidt, L., Santurkar, S., Tsipras, D., Talwar, K., and M ˛adry, A. (2018). Adversarially robust generalization requires more data. arXiv preprint arXiv:1804.11285.

Tsipras, D., Santurkar, S., Engstrom, L., Turner, A., and Madry, A. (2018). There is no free lunch in adversarial robustness (but there are unexpected benefits). arXiv preprint arXiv:1805.12152.

---

### Meta-Review · Area_Chair1 · 2018-12-14
**Interesting observation and results**

**Confidence:** 4
**Recommendation:** Accept (Poster)

**Metareview:**

This paper studies an interesting phenomenon related to adversarial training -- that adversarial robustness is quite sensitive to semantically lossless shifts in input data distribution.

Strengths
- Characterizes a previously unobserved phenomenon in adversarial training, which is quite relevant to ongoing research in the area.
- Interesting and novel theoretical analysis that motivates the relationship between adversarial robustness and the shape of input distribution.

Weaknesses
- Reviewers pointed out some shortcomings in experiments, and analysis of causes and remedies to adversarial robustness. The authors agree that given the current state of understanding, these are hard questions to pose good answers for. The result and observations by themselves are interesting and useful for the community.

The weakness that the paper does not propose a solution for the observed phenomenon remains, but all reviewers agree that the observation in itself is interesting. Therefore, I recommend that the paper be accepted.